# Reproductive Health in Women with Major β-Thalassemia: Evaluating Ovarian Reserve and Endocrine Complications

**DOI:** 10.3390/metabo14120717

**Published:** 2024-12-20

**Authors:** Vasileios Tsilionis, Efthalia Moustakli, Stefanos Dafopoulos, Athanasios Zikopoulos, Sotirios Sotiriou, Athanasios Zachariou, Konstantinos Dafopoulos

**Affiliations:** 1Department of Obstetrics and Gynecology, Faculty of Medicine, School of Health Sciences, University of Thessaly, 41110 Larissa, Greece; vtsilionis1983@gmail.com; 2Laboratory of Medical Genetics, Faculty of Medicine, School of Health Sciences, University of Ioannina, 45110 Ioannina, Greece; thaleia.moustakli@gmail.com; 3Department of Health Sciences, European University Cyprus, Nicosia 2404, Cyprus; stefanosntf2001@gmail.com; 4Obstetrics and Gynecology, Royal Devon and Exeter Hospital Barrack Rd., Exeter EX2 5DW, UK; thanzik92@gmail.com; 5Department of Embryology, Faculty of Medicine, School of Health Sciences, University of Thessaly, 41110 Larissa, Greece; sotiriousot@yahoo.gr; 6Department of Urology, School of Medicine, Ioannina University, 45110 Ioannina, Greece; zahari-ou@otenet.gr; 7IVF Unit, Department of Obstetrics and Gynecology, Faculty of Medicine, School of Health Sciences, University of Thessaly, 41110 Larissa, Greece

**Keywords:** hemoglobinopathies, HPG axis, reproductive endocrinology, ovarian biomarkers, reproductive health, infertility

## Abstract

Thalassemia is an autosomal recessive hereditary chronic hemolytic anemia characterized by a partial or complete deficiency in the synthesis of alpha- or beta-globin chains, which are essential components of adult hemoglobin. Mutations in the globin genes lead to the production of unstable globin chains that precipitate within cells, causing hemolysis. This shortens the lifespan of mature red blood cells (RBCs) and results in the premature destruction of RBC precursors in the bone marrow. Regular red blood cell transfusions are the standard treatment for thalassemia. However, these transfusions can lead to increased iron overload, which can impair vital systems such as the liver, heart, ovaries, and endocrine system. Focusing on female reproductive endocrinology, recurrent blood transfusions can cause iron accumulation in the pituitary and hypothalamus, leading to hypogonadotropic hypogonadism (HH), the most common endocrinopathy in these patients, affecting 40–91% of women. Recurrent transfusions and the resulting iron overload can also lead to oxidative stress and ovarian damage in patients with beta-thalassemia major (BTM). Despite advancements in iron chelation therapy, hypothalamic–pituitary damage associated with HH contributes to subfertility and sexual dysfunction, often with little to no recovery. In women exposed to gonadotoxic drugs, particularly those with BTM, anti-Mullerian hormone (AMH)—a marker of ovarian reserve—is frequently used to assess ovarian damage. This review aims to explore the pathophysiology of β-thalassemia and its major clinical manifestations, with a focus on endocrine complications and their impact on ovarian reserve. It also investigates how metabolomics can provide insights into the disease’s metabolic alterations and inform current and emerging therapeutic strategies to mitigate complications and optimize patient outcomes, potentially leading to more effective and personalized treatments.

## 1. Introduction

Globally, beta-thalassemia is the most common major monogenic hemoglobin disorder, affecting millions of individuals, particularly in regions such as the Mediterranean, Middle East, and Southeast Asia [1]. This hereditary condition results from insufficient (β+) or absent (β0) β-globin chain synthesis, leading to severe anemia due to ineffective erythropoiesis and hemolysis. The disease is further complicated by iron overload, which arises from frequent blood transfusions and ineffective erythropoiesis, resulting in oxidative stress that damages cellular components and tissues [2]. Advances in iron chelation therapy and regular blood transfusions have transformed this previously life-threatening condition, significantly lowering morbidity and mortality rates [3,4]. However, patients undergoing multiple transfusions may develop an imbalance between pro-oxidants and antioxidants, leading to excessive free radical production and increased reactive oxygen species (ROS) [5].

Among thalassemia patients, hypogonadotropic hypogonadism (HH) is the most prevalent endocrinopathy, affecting 40–91% of individuals. This condition arises from iron deposition (hemosiderosis) in the hypothalamus and pituitary gland, often due to repeated blood transfusions, and results in infertility and sexual dysfunction [6]. Despite advancements in chelation therapies that have improved outcomes, damage to the hypothalamic–pituitary axis is frequently irreversible [7].

Reproductive health in women with BTM is further complicated by uncertainties regarding ovarian reserve preservation. Granulosa cells in preantral and antral follicles secrete the anti-Müllerian hormone (AMH), a reliable biochemical marker of ovarian reserve that remains stable throughout the menstrual cycle. Researchers often use AMH to assess ovarian damage from gonadotoxic drugs, particularly in women who have undergone chemotherapy for cancer [8]. Despite frequent blood transfusions and complications from hemosiderosis, it is still uncertain whether ovarian reserve is preserved in patients with BTM [9].

This review aims to provide a comprehensive understanding of how BTM affects ovarian reserve, exploring the mechanisms of damage, diagnostic methods, and treatment approaches. By addressing these issues, we intend to help women with BTM regain their reproductive health and quality of life while providing advice on how to maintain fertility and fulfill the goals of this expanding patient group.

## 2. Endocrine Impact

The complex pathology of β-thalassemia major (BTH) is compounded by Thalassemic Endocrine Disease (TED), which encompasses multiple endocrine disorders that can arise from a young age [10]. Conditions like thalassemia bone disease (TBD) may worsen as patients age. Reproductive health, including fertility profiles, conception challenges, and pregnancy outcomes, is a significant aspect of TED. As assisted reproductive technologies (ART) and genetic counseling advance alongside BTH management, addressing TED necessitates specialized and vigilant care [11].

Genotype–phenotype correlations indicate that homozygous β-thalassemia patients are susceptible to iron overload (IO) in multiple organs, particularly the pancreas and heart, although predicting TED’s development remains challenging [11]. Studies attempting to model hormonal imbalances have not consistently linked factors such as IO, serum ferritin levels, organ iron deposits, chelation history, and splenectomy status to the severity of endocrine dysfunction [12]. Regardless of age, effective disease management is the most reliable predictor of TED risk, highlighting the sensitivity of endocrine glands to IO. Notably, one in four patients under age 12 already has at least one endocrine disorder, demonstrating TED’s early onset [13].

Endocrine complications associated with β-thalassemia include disruptions in the Growth Hormone/Insulin-like Growth Factor-1 (GH/IGF-1) axis, potentially leading to growth retardation and GH deficiency in both children and adults. Other common issues include hypogonadism, hypothyroidism, glucose metabolism disorders, hypoparathyroidism, adrenal insufficiency, and reduced bone mineral density (BMD) [14]. TED may also lead to secondary complications, with abnormalities in the cardio–metabolic profile increasing the risk of diabetes mellitus (DM) and related viral and dermatologic issues. Hypogonadism, which can have central or gonadal origins, presents in various forms, including low BMD, sexual dysfunction, delayed puberty, and fertility issues in both men and women [15,16].

Some studies have reported malnutrition in 70% of children under 12, with 23.33% experiencing at least one endocrine disorder. Higher ferritin levels and non-adherence to treatment correlate with endocrine disorders. One study found that 40% of 612 participants had endocrine disorders, primarily due to diabetes mellitus (DM) and osteoporosis, while another reported an 86.6% prevalence in 713 individuals. A meta-analysis indicated a 43.92% prevalence of endocrine disorders unrelated to DM [11,17].

TED significantly impacts healthcare costs and quality of life. Treatments targeting iron overload in BTH patients may improve endocrine outcomes, though evidence linking specific BTH therapies to better endocrine results has been inconsistent [18]. In a study of 31 transfusion-dependent BTH patients (average age 16.9 ± 3.8 years), follow-up over 5.9 ± 2.02 years showed that chelation therapy with deferasirox reduced endocrine involvement from 83% to 25.8% (*p* < 0.005). In a major multicenter study involving 426 transfusion-dependent thalassemia patients (median follow-up of 8 years), the results indicated that 121 participants had at least one endocrine disorder, while 187 had at least two. During follow-up, 104 new endocrine disorders were reported, reflecting a 9.7% risk (95% CI 6.3–13.1) of developing an additional disorder within five years. Baseline endocrine disorder count, age (*p* = 0.005), and TSH levels (*p* < 0.001) were significant predictors of new disorder onset, with extended deferasirox therapy appearing to reduce this risk [11].

## 3. Hypogonadism in BTM

Hypogonadism in transfusion-dependent thalassemia patients is often linked to iron overload, general health complications, disruptions in the hypothalamic–pituitary–gonadal (HPG) axis, and treatment side effects [7]. In females, primary or secondary amenorrhea can occur during puberty or adulthood, depending on the severity of endocrine disruption. The prevalence of hypogonadism varies across studies, particularly concerning pubertal status. Alongside cardiovascular risks from cardiac IO, prolonged hypogonadism significantly increases cardiovascular disease risk. Studies have shown a negative correlation between serum ferritin levels and pituitary gland volume, with one reporting a 23.6% prevalence of hypogonadism in patients of pubertal age. Additionally, some researchers suggest serum ferritin may predict delayed or failed puberty [19].

A cross-sectional study at a single tertiary center involving 58 transfusion-dependent thalassemia patients (33 males) aged 17–19 years found that 72.4% experienced normal puberty or delayed onset, while 26.7% had arrested puberty, necessitating hormonal treatment. Multivariate regression analysis indicated that serum ferritin was significantly linked to pubertal failure or arrest, with an odds ratio (OR) of 1.005 (95% CI 1.001–1.009; *p* = 0.028). Another study of 117 thalassemia patients indicated that iron overload negatively impacts sexual function [20,21].

In males, hypogonadism leads to impaired fertility and affects sperm parameters, potentially due to IO’s impact on Leydig cells and testosterone production. Similar effects are observed in females, where hypothalamic damage can disrupt hormone production and follicle maturation [22]. In a study of 54 women with BTM, AMH levels correlated with increased LH and reduced FSH, suggesting that significant ovarian reserve loss can occur before puberty. Another investigation of 61 female patients revealed reduced AMH levels among those with hypogonadism compared to those with normal hormonal profiles, suggesting that ovarian reserve declines earlier than previously thought in BTM [23].

## 4. BTH-Related Hypothyroidism

The prevalence of thyroid dysfunction in transfusion-dependent thalassemia (TDT), encompassing both primary and secondary hypothyroidism (clinical and subclinical), varies based on research methods, demographics, disease management, and specific medications. Thyroid function issues are often detected quickly, making routine ultrasound screening unnecessary. Patients who have undergone transplants or those with transfusion-dependent beta-thalassemia are also at risk for thyroid disease, though abnormalities may improve with effective chelation therapy [24].

In transfusion-dependent β-thalassemia major (BTM), iron overload can lead to heart and liver complications, endocrine dysfunction, and reproductive organ impairment [11]. Insufficient chelation therapy during childhood can result in gonadal dysfunction, which is prevalent in BTM and includes infertility, hypogonadotropic hypogonadism, delayed or absent sexual development, and osteoporosis [25]. The oxidative stress and dysfunction of the hypothalamic–pituitary-ovarian (HPO) axis caused by excessive iron deposits in endocrine organs contribute to infertility in BTM patients. Iron-induced hypogonadism is the most common reproductive issue, affecting 30–70% of patients worldwide. While transfusion-related endocrine disorders are prevalent, early diagnosis and improved treatments have significantly enhanced the quality of life and reproductive potential for BTM patients in recent decades [26].

## 5. Iron and Pathophysiology of BTM Infertility

The leading hypothesis for infertility in women with β-thalassemia major (BTM) posits that iron, whether directly or indirectly, affects the female reproductive system and the hypothalamic–pituitary–ovarian axis. Direct evidence indicates that iron’s indirect effects are primarily linked to iron-induced oxidative stress, while indirect effects likely stem from iron deposition in the reproductive system and the hypothalamic–pituitary axis [27]. Limited information exists regarding the pathophysiology of iron-induced infertility, and it remains unclear if the direct or indirect effects of iron are being adequately considered. The hemochromatosis model provides a suitable framework for exploring these effects on fertility in women with BTM. It is also important to differentiate between the hypothalamic pituitary axis and the female reproductive system as potential sites of iron’s influence on fertility [28] (Figure 1).

## 6. Oxidative Stress and Female Fertility

In patients with beta-thalassemia major (BTM), recurrent blood transfusions lead to iron overload and an accumulation of non-transferrin-bound iron (NTBI), which is particularly harmful to tissues. This condition is exacerbated by the downregulation of hepcidin, the principal regulator of systemic iron homeostasis, which enhances duodenal iron absorption [29]. Excess iron triggers the Fenton reaction, producing ROS and disturbing cellular redox equilibrium, resulting in chronic oxidative stress (OS). Oxidative stress stimulates lipid peroxidation of unsaturated fatty acids in cellular and organelle membranes, producing harmful byproducts such as 4-hydroxy-2′-nonenal and malondialdehyde (MDA). These byproducts disrupt protein synthesis, damage DNA, and impair cellular function [30]. Additionally, the release of heme and iron from excess unpaired alpha-globin chains increases the oxidant susceptibility of thalassemia red blood cells (RBCs), initiating self-amplifying redox reactions that further oxidize hemoglobin, accelerate RBC destruction, and deplete cellular reducing potential, such as glutathione (GSH) [31].

Homeostasis relies on a balance between oxidative and antioxidant processes. Oxidative stress arises when this balance is disrupted, leading to an excess of ROS. While ROS generation is a normal biological process in respiring cells, unchecked production can harm small antioxidant molecules and cellular macromolecules [32]. To mitigate ROS’s harmful effects, the body employs various enzymatic and non-enzymatic defense mechanisms. Enzymatic systems such as superoxide dismutase (SOD), catalase (CAT), and glutathione peroxidase (GPx) can neutralize ROS directly or sequentially. Non-enzymatic defenses include scavenging molecules obtained from the diet, such as vitamins C and E, and those synthesized endogenously, like glutathione, ubiquinols, and uric acid. These antioxidant nutrients undergo active recycling within various cellular compartments [33]. Hormonal influences may also affect antioxidant systems that protect against peroxidative damage. Antioxidant enzymes require specific microelements in their active centers, and the coordinated action of non-enzymatic antioxidants enhances their scavenging activity. Diet also influences the antioxidant system, relying on an adequate intake of vitamins and microelements for effective function against excess free radicals [34].

Data suggest that BTM patients experience a significant pro-oxidant/antioxidant imbalance, resulting in elevated oxidative stress. This imbalance is primarily due to tissue damage from secondary iron overload, leading to excessive free radical production, altered serum trace element levels, and changes in antioxidant enzyme activity [27]. Wassem et al. found that BTM patients had significantly lower levels of vitamin E, GPx, and SOD compared to controls, indicating increased oxidative stress in thalassemia [31]. Another study noted that children with BTM had higher levels of plasma thiobarbituric acid (TBA)-reactive substances, along with increased SOD activity and decreased CAT activity [35]. In the same study, serum ferritin was positively correlated with TBA-reactive compounds and SOD activity, suggesting a role for iron overload in oxidative stress within cells [36]. Additionally, significant decreases in total antioxidant capacity, vitamin E, and total iron-binding capacity were reported, alongside marked increases in lipid peroxides and iron levels [37]. These findings highlight the role of diminished antioxidant defenses and oxidative stress in the pathogenesis of BTM. Livrea et al. proposed measuring peroxidation products alongside antioxidant levels as a simple marker of iron toxicity in thalassemia, finding that concentrations of protein carbonyls, lipoperoxides (measured as MDA/TBA adducts), and conjugated diene lipid hydroperoxides (CD) increased by approximately threefold compared to the controls [37]. There was also a significant decrease in levels of lycopene, beta-carotene, ascorbate, vitamin E, and vitamin A. Blood levels of lycopene, vitamin A, and vitamin E were negatively correlated with transaminase levels, while ferritin and vitamin E exhibited an inverse relationship, indicating substantial consumption of this antioxidant during iron overload. The study indicated that lipid-soluble antioxidants might be significantly depleted due to iron-induced liver damage in thalassemia [38]. According to Chiou et al., BTM patients show markedly increased lipid peroxidation levels and significantly reduced plasma concentrations of vitamins A, E, and C. Thus, antioxidant therapy may be necessary to mitigate oxidative stress. They also found that while HCV infection did not significantly affect GSH levels or antioxidant enzyme activities, it contributed to the depletion of plasma vitamins E and C in BTM patients [38].

Although direct research linking iron-induced oxidative stress to infertility in women with BTM is lacking, existing studies on non-BTM patients suggest correlations between oxidative stress—though not specifically attributed to iron overload—and various physiological processes in the female reproductive system [27]. There is no published evidence directly connecting iron-induced oxidative stress to specific alterations in the hypothalamic–pituitary–gonadal (HPG) axis signaling pathway that could contribute to infertility. Therefore, the hypothesis that iron-induced oxidative stress is a primary factor in infertility among women with BTM is only indirectly supported by current evidence. This may occur not only through iron deposition and oxidative stress in the hypothalamus, pituitary, and female reproductive system but also through its effects on the liver and pancreas, leading to reduced hormone levels and serum antioxidant metabolism [27].

ROS have been shown to play a significant role in the pathophysiology of female infertility and normal reproductive system function. Oxidative stress, arising from an imbalance between ROS production and the antioxidants’ ability to neutralize them, affects both natural and assisted fertility [39]. ROS influence various physiological processes, including oocyte maturation, fertilization, embryo development, and pregnancy [40]. Current data suggest that oxidative stress and dietary antioxidants may impact the initiation and maintenance of a viable pregnancy [41]. Physiological levels of ROS regulate several aspects of folliculogenesis, oocyte maturation, the endometrial cycle, luteolysis, implantation, embryogenesis, and pregnancy through various signaling pathways [42]. Healthy and fertile ova depend on proper ovarian function, which is contingent on the continuous growth and maintenance of ovarian follicles. The effects of oxidative stress on primordial, developing, and preovulatory follicles, oocytes, and early embryos have been well studied, demonstrating the impact on cell types and molecular targets [43]. Regression of the corpus luteum (CL) occurs near the end of the non-fertile cycle and is mediated by ROS-induced apoptotic cell death [44].

Emerging research indicates that oxidative stress may influence a range of reproductive disorders, including endometriosis, preeclampsia, hydatidiform moles, preterm labor, spontaneous abortions, polycystic ovary syndrome (PCOS), recurrent miscarriages, and intrauterine growth retardation [42]. Thus, the significance of oxidative stress in female reproduction is becoming increasingly recognized. A recent study by Agarwal et al. explored the pathophysiology of reproductive disorders associated with oxidative stress and elucidated the mechanisms of redox cell signaling, assessing its impact on physiological and pathological processes within the female reproductive system [45].

Iron excess has been noted in various compartments of the peritoneal cavity—such as peritoneal fluid, endometriotic lesions, the peritoneum, and macrophages—in patients with endometriosis, warranting particular attention. This excess iron affects several processes involved in the pathogenesis of endometriosis [46]. Infertility associated with endometriosis may be influenced by iron-induced oxidative stress, which modulates the expression of genes encoding immunoregulators, cytokines, and cell adhesion molecules critical to the condition’s pathophysiology [47]. A recent study also investigated the relationship between oxidative stress, the underlying causes of infertility, preovulatory ovarian hormones, and the ovarian response to gonadotropin stimulation in patients undergoing assisted reproductive technologies (ART). The overall cause of infertility was not significantly correlated with either the total antioxidant capacity (TAC) of the plasma or follicular fluid (FF). However, a statistically significant positive correlation was found between FF E(2) levels and TAC. Oxidative stress impacts the granulosa cell’s steroid hormone synthesis, particularly E(2), a key indicator of the ovarian response [48]. Dysregulation in granulosa cell steroid hormone synthesis, including E(2), may contribute to the decreased ovarian response in women with BTM [40]. Furthermore, oxidative stress has been hypothesized to mediate the age-related decline in fertility. Recent research suggests that elevated oxidative stress exacerbates ovarian aging, potentially linked to the accumulation of nonheme iron in ovarian stromal tissue [49]. This issue is particularly relevant for our patient population, which experiences age-related fertility decline due to an older average age, a result of advancements in chelation therapy techniques. Another study investigated the impact of retinol and alpha-tocopherol—two naturally occurring antioxidants that inhibit lipid peroxidation and protect cells from oxidative stress-induced damage. The results revealed a correlation between reduced levels of these antioxidants and both abnormal semen parameters in men and anovulation in women [50].

## 7. Evaluation of Ovarian Reserve in Women with BTM

Ovarian reserve assessment is crucial for assessing reproductive potential and detecting ovarian dysfunction or failure early in women with beta-thalassemia major (BTM) [51]. Endocrine complications and chronic iron overload from several blood transfusions frequently impede ovarian function in women with BTM. A comprehensive evaluation of ovarian reserve is necessary to guide clinical decisions on hormone replacement therapy (HRT), fertility preservation, and overall reproductive health care [52].

One of the primary biomarkers used to assess ovarian reserve is the anti-Müllerian hormone (AMH), which is thought to be a reliable indication of ovarian function. By reflecting the remaining follicular pool, this hormone acts as a gauge of the remaining reproductive capacity [8]. Women with BTM often have lower AMH levels than healthy women because of ovarian damage caused by iron overload. Even in cases where menstrual cycles are regular, this decline in AMH may indicate a declining ovarian reserve and be a predictor of early ovarian failure [51]. Since AMH may be assessed at any point during the menstrual cycle, unlike other biomarkers, it is a useful and trustworthy diagnostic for monitoring ovarian reserve in this population [8].

The measurement of follicle-stimulating hormone (FSH) levels serves as a critical biomarker for assessing reproductive potential and ovarian reserve, offering insights into the physiological demand for the activation of ovarian follicles, especially during the early follicular phase of the menstrual cycle. Elevated FSH levels indicate a reduced ovarian reserve, a sign that the ovaries are struggling to respond to hormonal cues [53]. Hypogonadotropic hypogonadism, often seen in beta-thalassemia major (BTM) patients due to iron-induced pituitary damage, is instead characterized by low FSH and LH levels, reflecting the impaired pituitary production of these hormones necessary for normal ovarian function. Chronic elevations in FSH may indicate compromised ovarian function, particularly when combined with other measures of ovarian reserve [7].

In addition to FSH, estradiol (E2) levels are often used in evaluating ovarian reserve. Estradiol, produced by developing ovarian follicles, is an indirect indicator of ovarian activity. Low estradiol levels in BTM women may result from ovarian failure or insufficient pituitary stimulation caused by iron-induced damage [54]. Elevated FSH levels, accompanied by elevated estradiol levels, may indicate poor ovarian response or diminished ovarian reserve. However, because estradiol levels vary with the menstrual cycle, its diagnostic value is limited when used alone. Estradiol is typically assessed alongside FSH during the early follicular phase to provide a more comprehensive view of ovarian function [55].

Ultrasound examinations can directly visualize ovarian reserve, particularly in relation to the antral follicle count (AFC). Transvaginal ultrasound is used to perform AFC during the early follicular phase, which counts the number of small antral follicles in the ovaries. Reduced AFC is a symptom of poor ovarian function and may be caused by iron-related ovarian damage in women with BTM [55]. This method works well for predicting potential fertility and guiding treatment decisions. Further information can also be obtained by measuring ovarian volume, another characteristic that can be analyzed with ultrasound, since smaller ovaries may indicate atrophy or lower follicular activity [56].

Endocrine evaluation is crucial for women with BTM, particularly if iron excess affects the hypothalamic–pituitary-gonadal axis. A common disorder in this population is hypogonadotropic hypogonadism, which causes ovarian dysfunction by impairing the synthesis of FSH and luteinizing hormone (LH). To assess iron deposition, serum ferritin and T2* MRI are essential components of routine iron level monitoring in understanding the impact of iron excess on reproductive health. Effective iron chelation therapy is essential for long-term care for BTM patients because it can preserve fertility and lessen ovarian damage [57].

Ovarian reserve testing has important clinical ramifications for women with BTM. Timely fertility counseling and management are made possible by early identification of reduced ovarian reserves [55]. Oocyte or embryo cryopreservation is one fertility preservation treatment that may help women with diminished ovarian reserve, especially before further ovarian decline develops. In women facing ovarian failure, hormone replacement treatment (HRT) is frequently required to correct hypogonadism, induce puberty, regulate menstrual cycles, and maintain secondary sexual characteristics [58]. To guarantee that the right kind of medication is administered, it is crucial to differentiate between ovarian failure and the primary causes of amenorrhea [59].

Women with significantly reduced ovarian reserve may need to use assisted reproductive technologies (ART) in order to conceive. In these cases, ovarian reserve testing is essential as it aids in identifying the type and timing of ART, such as in vitro fertilization (IVF) or donor egg use. Clinical professionals use testing to help patients make reproductive decisions and as a foundation for customized treatment plans aimed at optimizing fertility results [60].

Despite its benefits, ovarian reserve testing has a number of disadvantages, particularly with regard to BTM. FSH and estradiol are two biomarkers that rely on the cycle and must be checked at specific periods; thus, women who experience amenorrhea or irregular cycles may find it challenging to test for these [61]. The lack of normative information specific to women with BTM makes it even more difficult to define what “normal” ovarian reserve is in this cohort. Since the majority of biomarker reference ranges are based on healthy women and might not entirely apply to people with BTM, data interpretation can be challenging [51]. Furthermore, the outcomes of treatment plans, like iron chelation programs, can vary from one BTM patient to another, making it difficult to generalize findings to the entire community. The varying effects of transfusion frequencies and chelation methods on ovarian reserve can complicate the evaluation process [62].

For women with β-thalassemia major, ovarian reserve assessment is essential for guiding fertility and hormonal management strategies, as well as the early identification of reproductive problems [27]. AMH, FSH, estradiol, and AFC are biomarkers that must be regularly evaluated in order to observe ovarian function, particularly in the presence of iron overload and endocrine issues. Although standardizing these tests for BTM women is challenging, early evaluation can inform customized treatments aiming to preserve fertility and improve long-term reproductive outcomes (Table 1).

## 8. Fertility Preservation in Hemoglobinopathy Patients

Fertility preservation before hematopoietic stem cell transplantation (HSCT) is essential due to the risk of gonadal failure after HSCT. Early counseling is crucial, regardless of the patient’s age, to help them safeguard their reproductive potential [63]. Additionally, concerns about sterility and treatment toxicity are significant barriers to accepting HSCT. Financial issues, patient and family preferences, and provider awareness also continue to limit access to fertility preservation options [64].

The main factors in choosing a fertility preservation method for a female patient include whether she has experienced menarche and how urgently gonadotoxic treatment is needed. For patients with hemoglobinopathies, fertility preservation before HSCT is usually not an emergency [65]. This allows for better coordination and health optimization. Menarche may be delayed in some cases, allowing for options like oocyte cryopreservation and controlled ovarian hyperstimulation. For patients who have not yet reached menarche and cannot delay transplantation, ovarian tissue cryopreservation (OTC) is the only available option for fertility preservation due to age and disease severity [66].

For patients who have reached menarche, both OTC and oocyte cryopreservation are viable options. Embryo cryopreservation is also possible but is less common among children and young adults due to ethical concerns and the need for a sperm donor [67].

## 9. Oocyte Cryopreservation

Controlled ovarian hyperstimulation (COH) requires ten to fourteen days of close monitoring and medication [68]. Hormone levels are checked through routine blood tests, and follicle growth is tracked by transvaginal or transabdominal ultrasound. Then, under anesthesia, oocytes are extracted through a minimally invasive procedure [69].

Common side effects of COH include headache, nausea, abdominal bloating, and pain. Although less common, ovarian hyperstimulation syndrome (OHSS) can occur, leading to fluid accumulation (ascites), cardiopulmonary effusions, and venous thromboembolism (VTE) [70]. For patients with hemoglobinopathies, these side effects are particularly concerning due to their altered pain perception and lower tolerance for COH discomfort. Those with vascular, lung, or kidney issues are at even higher risk from OHSS, as they may not handle fluid shifts as well [71]. In patients with sickle cell disease (SCD), COH raises the risk of vaso-occlusive crises (VOC) and acute chest syndrome (ACS). To date, four cases of acute pain crises during COH have been documented [72].

In SCD patients, the number of oocytes retrieved through COH can vary from four to thirty-one. Research suggests that harvesting at least fifteen oocytes is ideal for a single cycle to maximize pregnancy chances. However, only 25% of the patients reached this target [73]. Financial and scheduling constraints may explain why none of these patients underwent repeated cycles to preserve fertility. There are limited publicly available data on COH procedures and outcomes for adolescents with and without hemoglobinopathies [74]. Adult dosing regimens, which consider age, FSH level, and AFC, are typically used as a reference, though they may require significant adjustments. For example, Lavery et al. found that in 80% of cases, dose adjustments were needed in their cohort of eight adolescent girls [75].

In untransplanted patients with beta-thalassemia major (BTM), there have been several reports of successful, uncomplicated IVF cycles and ovulation induction. However, no COH procedures for BTM before fertility preservation have been documented [76,77]. Since iron chelators are contraindicated during pregnancy, the authors of these studies suggest stopping them before ovulation induction. However, this is not necessary solely to preserve fertility [77].

## 10. Ovarian Tissue Cryopreservation

One fertility preservation technique (FPT) that is becoming more widely used is ovarian tissue cryopreservation (OTC). The American Society for Reproductive Medicine (ASRM) no longer considers OTC to be experimental as of 2019, though this may differ by country. As of 2017, OTC had resulted in 130 live births [78], and by 2020, that number was expected to reach about 200 [79]. OTC may be the only available FPT option for young hemoglobinopathy patients who need to undergo HSCT.

The most common method for OTC involves an outpatient laparoscopic procedure to remove one ovary or part of it. This tissue can be stored and later used for ovarian tissue transplantation (OTT). OTC preserves a high number of primordial follicles, which may restore ovarian endocrine function after OTT. According to a meta-analysis of 309 OTT cases, 64% experienced endocrine restoration, evidenced by regular menstrual cycles, ovarian follicle growth on ultrasound, or pregnancy. The clinical pregnancy rate after OTT was 57.5% [80].

In the limited cases of post-OTT hemoglobinopathy patients, the recovery of ovarian endocrine function has been documented. In 2006, Donnez et al. first reported ovarian function restoration after orthotopic transplantation in a patient with HbSS. Before HSCT, at age 21, she underwent OTC. Following her transplant, she required hormone supplements and later had an OTT, which led to regular menstruation, ovarian function, and follicle development [81]. These findings have since been confirmed in other adolescent hemoglobinopathy patients, including those who were prepubertal [82].

The first documented OTC pregnancy from a prepubertal patient was published by Mamsen et al. in 2021. At age nine, a girl with beta-thalassemia underwent OTC before HSCT. She later recovered ovarian function and conceived through IVF at age 23 following OTT. Another patient required three OTTs and fourteen rounds of IVF to achieve a live birth [83]. While spontaneous pregnancy is possible after OTC, most cases require in vitro fertilization (IVF). The first live birth from cryopreserved ovarian tissue from a premenarchal pubertal female was reported by Demeestere et al. in 2015 [84]. OTC was performed at age 13 for a patient with SCD who, later, upon wanting to conceive, was diagnosed with primary ovarian insufficiency. However, two years after OTT, she achieved a spontaneous pregnancy. Unfortunately, most studies do not provide enough details about IVF indications or the number of cycles needed to estimate the likelihood of spontaneous conception after OTT, nor do they report on infertility rates in OTT patients. Additionally, patients’ reactions to HSCT vary widely; some recover fully, while others do not. Since untreated SCD patients have higher risks of stillbirth and fetal growth restriction, different pregnancy outcomes are expected in those who are not cured [85].

## 11. Other Fertility Considerations

Women with SCD and beta-thalassemia are at higher risk for obstetric issues, including maternal mortality, intrauterine fetal death, preeclampsia, preterm birth, and spontaneous miscarriage. Hypercoagulability, a common comorbidity in various hemoglobinopathies, is one factor that increases these risks. In addition to providing appropriate preconception counseling, maternal-fetal medicine specialists should consider discussing surrogacy as an option for women with significant comorbidities who wish to have biological children [73].

Genetic testing is also a critical aspect of reproductive care. To reduce the risk of having an affected child, preimplantation genetic testing (PGT) should be offered to individuals with hemoglobinopathy mutations. Providing education on PGT and surrogacy can help patients and their families understand the full range of fertility options, even if they are not immediately ready for parenthood after preserving their fertility.

## 12. Clinical Challenges and Limitations

Due to the chronic nature of the disease and the impact of long-term therapies, especially iron overload from many blood transfusions, women with β-thalassemia major have considerable reproductive and endocrine issues [86]. Menstruation disruption, significant fertility impairment, and a host of other issues impacting both short- and long-term reproductive results can result from iron buildup in important endocrine organs. Despite advancements in treatment, there are numerous challenges to overcome in the diagnosis and management of these issues [87]. These challenges include the inability to monitor iron levels, the diversity of patient reactions to medications, the absence of defined protocols, and the restricted alternatives for fertility preservation. These problems underscore the pressing need for multidisciplinary approaches and additional research to optimize care, especially when combined with the psychosocial effects and high-risk nature of pregnancies among thalassemia patients (Table 2).

## 13. Conclusions

This review elucidates the complex interplay between oxidative stress, iron overload, and reproductive health in women with significant β-thalassemia (BTM). Frequent blood transfusions for BTM management lead to substantial iron accumulation, exacerbating oxidative stress and disrupting the balance between oxidative and antioxidative processes [27]. This biochemical imbalance significantly impairs fertility, adversely affecting ovarian reserve and endocrine function. The interference of ROS with cellular processes and signaling pathways in the hypothalamic–pituitary–gonadal (HPG) axis further underscores the need for targeted approaches to reduce oxidative damage [94]. Additionally, while direct links between iron-induced oxidative stress and infertility in BTM have not been thoroughly explored, indirect evidence suggests that excessive iron deposition may disrupt hormonal balances and metabolic processes, thereby affecting reproductive health [95].

For women with severe β-thalassemia, antioxidant treatment along with frequent iron level monitoring may improve reproductive outcomes and safeguard overall health. In people with β-thalassemia, the results show that oxidative stress has a detrimental effect on important reproductive processes like oocyte maturation, embryo development, and hormone control [96]. Therefore, further research is needed to explore the mechanisms underlying reproductive dysfunction in this population and to develop comprehensive care strategies that integrate hematological and reproductive health considerations.

According to this review’s key insights, a multidisciplinary strategy that incorporates hematological and reproductive health issues is crucial in addressing reproductive failure in women with BTM. Future studies should concentrate on developing fertility preservation procedures specific to this patient group and investigating new antioxidants to reduce oxidative damage. Optimizing reproductive results will also require exploring cutting-edge methods to track ovarian reserve and improve chelation treatments. By filling in these gaps, we can promote all-encompassing care approaches that improve BTM women’s reproductive potential and quality of life.

## Figures and Tables

**Figure 1 metabolites-14-00717-f001:**
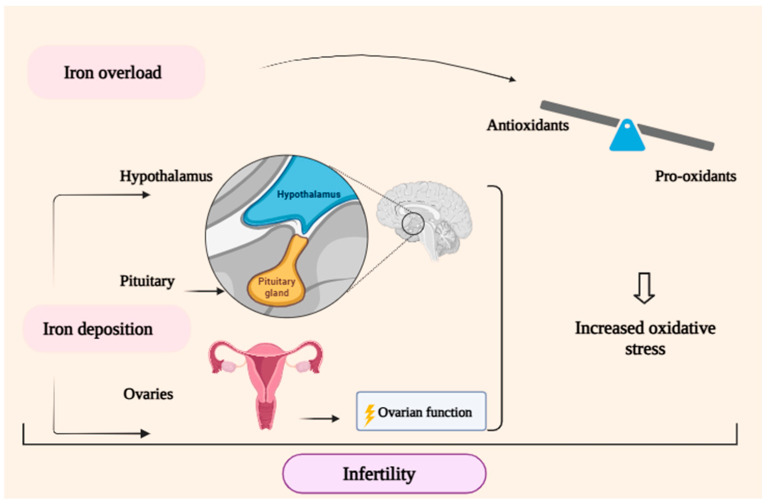
The primary cause of infertility in female patients with β-thalassemia major (BTM) seems to be attributable to both direct and indirect effects of iron overload. Dysfunction in the female reproductive axis arises from the synergistic impacts of increased oxidative stress and iron deposition, which result from a significant imbalance between pro-oxidants and antioxidants.

**Table 1 metabolites-14-00717-t001:** This table outlines the main biomarkers and diagnostic instruments that are used to assess the reproductive health and ovarian reserves in women with β-thalassemia major. Each biomarker is described, along with its clinical significance and particular limitations that are pertinent to the BTM population. Hormonal markers (estradiol, FSH, and AMH), imaging tests (ovarian volume, AFC), iron overload measurements, and endocrine conditions such as hypogonadotropic hypogonadism (HH) are among the biomarkers. Limitations include things like measurement variability, iron overload effects, and cycle dependence that can make diagnosis and treatment challenging.

Biomarker	Description	Clinical Significance	Limitations	References
Anti-Müllerian Hormone (AMH)	Hormone representing ovarian reserve Released by small antral and pre-antral follicles	Direct indicator of ovarian reserve Low levels suggest early ovarian failure or reduced ovarian reserve	Influenced by ovarian damage from iron overload No cycle dependence and no established limits for BTM	(La Marca et al., 2014; Broer et al., 2014) [63,64]
Follicle-Stimulating Hormone (FSH)	Hormone promoting follicular growth. Measured during early follicular phase	Elevated FSH indicates reduced ovarian reserve or ovarian failure	Cycle-dependent Highly variable Increased when pituitary injury causes HH	(Gougeon et al., 1996) [65]
Estradiol (E2)	Follicle growth releases estradiol sign of ovarian activity	Low levels are used in conjunction with FSH to assess ovarian reserve representing decreased ovarian function	Cycle-dependent Unreliable when used alone Affected by HH and other variables	(Fanchin et al., 2003) [66]
Antral Follicle Count (AFC)	Number of small follicles (2–10 mm) counted via ultrasound	Direct, visible depiction of ovarian reserve that aids in ART guidance and reproductive potential prediction	Influenced by iron-induced ovarian damage Requires expert ultrasound evaluation	(Dewailly et al., 2014) [67]
Ovarian Volume	Ultrasonography measurement of the ovaries’ size	Ovarian atrophy or decreased ovarian reserve may be indicated by a reduced ovarian volume	Less precise than AFC Actual functional capacity might not be reflected solely by ovarian size	(Kim et al., 2019) [68]
Iron Overload Assessment	T2* MRI and serum ferritin are used to measure the amount of iron in the heart, liver, and other organs	Elevated iron levels are linked to an increased risk of endocrine complications and ovarian damage	Requires constant observation and treatment with iron chelation	(Fanchi et al., 2003; Berdoukas et al., 2015) [69,70]
Hypogonadotropic Hypogonadism (HH)	Pituitary damage-related endocrine disease that impacts the synthesis of LH and FSH	To retain fertility and induce puberty, early identification enables prompt hormone replacement treatment (HRT)	Specialized endocrine examination is necessary to differentiate primary ovarian failure from central causes	(De Sanctis et al., 2018; De Sanctis et al., 2015) [16,71]

**Table 2 metabolites-14-00717-t002:** The primary obstacles to researching reproductive health in women with β-thalassemia major are shown in this table. It highlights topics including patient population heterogeneity, inconsistencies in ovarian reserve assessments, the absence of long-term data, and confounding variables like chelation therapy. The limitations of small sample sizes and the paucity of research on these patients’ options for fertility preservation are also highlighted.

**Challenges/Limitations**	**Descriptions**	**References**
Heterogeneity of patient populations	The formation of hypotheses across populations is complicated by the heterogenous disease severities observed among women with β-thalassemia. Different transfusion and chelation regimens result in inconsistent iron overload management, which impacts ovarian reserve and endocrine health outcomes.	(Betts et al., 2020) [88]
Limited longitudinal data	Most studies provide cross-sectional data and do not follow endocrine and reproductive health from childhood into adulthood. The effect of early therapies on long-term ovarian reserve and endocrine function is not well captured in many studies comparing early disease management to late-onset complications.	(Moolhuijsen et al., 2020) [8]
Inconsistent measurement of ovarian reserve	AMH, FSH, and AFC are only a few biomarkers used in different research, which makes comparing results more difficult. Without guidelines for ovarian reserve markers in patients with β-thalassemia, it is difficult to interpret results.	(Shahrokh et al., 2016) [89]
Cofounding factors	Research on the direct effects of β-thalassemia on reproductive health can be complicated by the effects of various chelation drugs. Direct attribution of ovarian dysfunction to β-thalassemia may be complicated by concurrent diseases.	(Amjad et al., 2020) [90]
Small sample size	The condition’s rarity limits statistical power since studies with β-thalassemia major typically have a small sample size. Results from small sample sizes may be biased toward more severe outcomes, rendering them less applicable to the larger β-thalassemia community.	(Button et al., 2013) [91]
Limited focus on fertility preservation	Little information is available regarding the efficacy of fertility preservation methods in β-thalassemia. Research on fertility interventions generally underrepresents women given the lack of access to ART and fertility preservation programs.	(Pawłowski et al., 2023; Roberts et al. 2024) [92,93]

## Data Availability

Data are unavailable due to privacy or ethical restrictions.

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
