# Peer review of "Reproductive Health in Women with Major β-Thalassemia: Evaluating Ovarian Reserve and Endocrine Complications"

_metabolites, 2024, doi:10.3390/metabo14120717_

Round 1

Reviewer 1 Report

Comments and Suggestions for Authors

ABSTRACT:

The abstract of this review appears to have poor alignment with the title. For example, the first 8 of 21 lines of the abstract has only introduction of beta-thalassemia and thus seem unnecessary. The abstract needs to condense/curtail the 1st 8 lines.

Main Text:

1.       The section (2. Pathophysiology of beta-thalassemia) appears again to be superfluous and irrelevant for a review of specific title as the author present here. This entire section can be deleted and only a couple of lines may be added to the introduction.  

2.       Page 9-10: TABLE (-No number or legend of the this table could be found).

In this table on 9-10, how did the authors interpret the significance and limitations of the biomarkers ? This table must include the references of the work that has been cited in it. A third column must be provided citing with corresponding references of the previous studies.

3.       Page 13: TABLE ( No number or legend of the this table could be found ??). This is again a major flaw in the presentation of this paper.

The table on page 13 is furnished by the authors to describe for example, that the heterogeneity of the B-thal patient population could be based on either hypotheses on heterogenous disease severities or transfusion and chelation regimens probably leading to iron overload management impacting the ovarian reserve and endocrine health outcomes in women with b-thal. Did the author evaluate these hypotheses? If yes, where is supporting information? If no, a third column must be provided citing with all the corresponding references.

Comments on the Quality of English Language

In addition, this manuscript needs several improvements in its overall organisation e.g. formatting and spelling etc. There are numerous corrections which make the review difficult.  

See a few examples: 

1.       Sickle cell disease (SCD): to be introduced once in the outset, and then the acronym SCD should be used all throughout. 

2.       Authors are advised to correct the following mistakes:

o   Numbering and spelling of conclusion section is wrong. see below 

o   13:  What is meant by the section heading Conclusionssssss ? The authors must put adequate attention and proofreading of the final version manuscript submission- to avoid such unacceptable typographical mistakes.

o   This manuscript elucidates… : correct to This 'review' elucidates..

Reviewer 2 Report

Comments and Suggestions for Authors

The manuscript is well-written, innovative and very needful. This type of review is very rare and will appreciate the authors for their attempt.  The only mistake i found is the spelling of the conclusion is not proper. 

Reviewer 3 Report

Comments and Suggestions for Authors

Dear Authors,

This review article we have written on reproductive health and endocrine complications in thalassemia is very important.

Because thalassemia patients now want to get married and have children at an adult age.

I would like to thank you for writing about this topic in full detail.

I congratulate the all team.

Best regards

Author Response

Plese see the attachment. 

Reviewer 4 Report

Comments and Suggestions for Authors

Major Comments

1.     Introduction

Page 2 line 48-55 The introduction could better frame the clinical important of study reproductive in women with BTM, highlight the global burden and of β-thalassemia and its implication for quality of life would improve context-setting(Alshalani, Li, Juffermans, Seghatchian, & Acker, 2019;  Tuo et al., 2024).

2.     Conclusion
Please reiterate the key takeaways, propose future research directions, such as, investigating the role of novel antioxidant or improving fertility preservation protocols

Minor Comments

1.      Language and grammar

Ensure consistent use of abbreviation ( e.g “BTM” for”β-thalassemia major throughout the text)

Page 1, Line 23, the full term “beta thalassemia major is not followed by the abbreviation BTM) on this section, even though it appears later in the manuscript.

Page 2, line 69 only BTM is needed here.

Page 4, line 20 only BTM is needed here.

Page 4. Line 183 -185 Remove the incomplete concision section 

2.     Table formatting

Page 10 Add an appropriate caption to the table for clarity

Ensure consistent font style and size throughout the table  

3.     Reference

Ensure all references are up-to-date, especially for the introduction, some reference provided for introduction listed as follow.

References

Alshalani, A., Li, W., Juffermans, N. P., Seghatchian, J., & Acker, J. P. (2019). Biological mechanisms implicated in adverse outcomes of sex mismatched transfusions. Transfusion and Apheresis Science, 58(3), 351-356.

Tuo, Y., Li, Y., Li, Y., Ma, J., Yang, X., Wu, S., . . . He, Z. (2024). Global, regional, and national burden of thalassemia, 1990–2021: a systematic analysis for the global burden of disease study 2021. eClinicalMedicine, 72. doi:10.1016/j.eclinm.2024.102619

Round 2

Reviewer 1 Report

Comments and Suggestions for Authors

A clean version of the manuscript (after removing all the track-changes, highlights and corrections following the revision comments) is requested.

Author Response

Comment

A clean version of the manuscript (after removing all the track-changes, highlights and corrections following the revision comments) is requested.

Response

Thank you for your feedback and the opportunity to revise our manuscript. As requested, we have prepared a clean version of the manuscript without track changes, but with the modified parts highlighted. This version has been included in our submission.

Round 3

Reviewer 1 Report

Comments and Suggestions for Authors

All comments are addressed well.

Comments on the Quality of English Language

Acceptable.